# Exploring the Venom Gland Transcriptome of *Bothrops asper* and *Bothrops jararaca*: De Novo Assembly and Analysis of Novel Toxic Proteins

**DOI:** 10.3390/toxins16120511

**Published:** 2024-11-27

**Authors:** Joseph Espín-Angulo, Doris Vela

**Affiliations:** 1Facultad de Ciencias Exactas y Naturales, Pontificia Universidad Católica del Ecuador, Quito 170525, Ecuador; 2Facultad de Ciencias Exactas y Naturales, Laboratorio de Genética Evolutiva, Pontificia Universidad Católica del Ecuador, Quito 170525, Ecuador

**Keywords:** arylsulfatase, bioinformatics tools, de novo assembly, dihydroorotate dehydrogenase

## Abstract

Previous proteomic studies of viperid venom revealed that it is mainly composed of metalloproteinases (SVMPs), serine proteinases (SVSPs), phospholipase A2 (PLA2), and C-type lectins (CTLs). However, other proteins appear in minor amounts that affect prey and need to be identified. This study aimed to identify novel toxic proteins in the venom gland transcriptome of *Bothrops asper* and *Bothrops jararaca*, using data from NCBI. Bioinformatics tools were used to assemble, identify, and compare potentially novel proteins in both species, and we performed functional annotation with BLASTX against the NR database. While previous assemblies have been performed for *B. jararaca*, this is the first assembly of the *B. asper* venom gland transcriptome. Proteins with potentially novel functions were identified, including arylsulfatase and dihydroorotate dehydrogenase, among others, that could have implications for venom toxicity. These results suggest that the identified proteins may contribute to venom toxic variation and provide new opportunities for antivenom research. The study improves the understanding of the protein composition of *Bothrops* venom and suggests new possibilities for the development of treatments and antivenoms.

## 1. Introduction

Throughout their evolutionary history, snakes have undergone notable changes, such as the development of loreal pits, the implementation of retractable fangs, and the emergence of venom glands [1]. Some species exhibiting these features belong to the families Atractaspidae, Elapidae, and Viperidae [2]. The latter family possesses one of the most complex and challenging proteomes to understand, as a wide range of proteins with various physiological activities interact within the venom, producing effects ranging from mild reactions like allergies and dermatitis to severe outcomes such as hemorrhage, necrosis, respiratory issues, and death [3,4]. However, the bioactive compounds involved in these effects have potential therapeutic applications in treating hypertension and congestive heart failure [5].

Snake venom composition can vary due to environmental conditions, sex, age, and diet [6,7]. However, species within the Viperidae and Crotalidae families present venoms rich in proteins that affect hemostasis, promoting antiplatelet effects, in contrast to Elapidae species, whose venoms predominantly exhibit neurotoxic effects [7,8]. In the Viperidae family, the genus *Bothrops* is one of the most diverse, with over 30 species and subspecies distributed across Central and South America [9]. Its venom has a rich protein composition, including phospholipases, hyaluronidases, metalloproteases, serine proteinases, and C-type lectins [10].

Among this genus, *Bothrops jararaca* stands out. It is endemic to the tropical and subtropical forests of southeastern Brazil, northeastern Paraguay, and northern Argentina [11], causing over 90% of snakebite incidents in the state of São Paulo [12]. Advances in high-throughput technologies have enabled the sequencing of venom glands, allowing for the study of their complex transcriptomic and proteomic compositions. Notable discoveries include the bradykinin potentiator factor (BPF), essential for the antihypertensive drug Captopril, and the anticoagulant Exanta [4].

In contrast, *Bothrops asper* is distributed across Mexico, Central, and South America and is responsible for 77.13% of bites in rural areas of Guayas and Los Ríos [13]. However, unlike *Bothrops jararaca*, transcriptomic and proteomic studies of *B. asper* are less extensive, focusing primarily on highly expressed toxins, leaving many venom proteins yet to be discovered [14,15].

In this study, we performed for the first time a de novo assembly of the venom gland transcriptome of *Bothrops asper*, which was compared with the de novo assembly of *Bothrops jararaca*. The main objective was to generate high-quality assemblies and identify new proteins, including their potential domains and conserved sites, based on data from other species such as insects, arachnids, reptiles, cnidarians, and bacteria.

## 2. Results and Discussion

### 2.1. De Novo Transcriptome Assembly and Quality

The transcriptome of the venom gland of *Bothrops jararaca* and *Bothrops asper* contains 16,574,152 and 13,295,995 paired reads, respectively (Table 1). The de novo assembly using Trinity generated 76,821 contigs (N50 = 1678) for *B. jararaca* and 54,771 contigs (N50 = 1453) for *B. asper*. Subsequently, the assembled transcriptome mapped against the raw reads, achieving an overall alignment of 89.01% for *B. asper* and 89.64% for *B. jararaca.*

The transcriptome completeness, assessed by Benchmarking Universal Single Copy Orthologs (BUSCO v5.5.0) with the eukaryota_odb10 lineage, showed that 89.4% of the genes are complete in *Bothrops jararaca* and 79.2% in *Bothrops asper* (Figure 1). On the other hand, using the sauropsida_odb10 lineage, we observed a decrease in the percentage of complete genes, with 50.2% for *B. jararaca* and 40.2% for *B. asper* (Figure 2).

This study addressed the analysis of the venom gland transcriptome of *Bothrops jararaca*, with a total of 76,821 contigs and an N50 length of 1678 bp, and *Bothrops asper*, with 54,771 contigs and an N50 length of 1453 bp. To date, venom gland transcriptome assemblies have been performed in the family Viperidae in species such as *Bothrops moojeni* [16], *Bothrops jararaca* [4], *Tropidolaemus wagleri* [17], *Azemiops feae* [18], and *Daboia siamensis* [19]. These studies show values comparable to those obtained in our research. For example, *Bothrops moojeni* identified 59,358 contigs with an N50 length of 894 bp, *Bothrops jararaca* yielded 76,765 contigs with an N50 length of 1104 bp, and *Tropidolaemus wagleri* found 58,914 contigs with an N50 length of 1032 bp. The assembly of *Bothrops jararaca* by Pereira and colleagues [4], differs from ours due to errors in K-mer size. Despite this issue, the assembly closely aligned with studies of other Viperidae species. According to the literature, the BUSCO quality of the assembly for *B. jararaca* is 85.8% gene complete, using the eukaryota_odb10 lineage [4]. In our study, using the same lineage, we achieved 89.4% complete genes for *B. jararaca* and 79.2% for *B. asper*. However, using a more specific lineage, such as sauropsida_odb10, we observed a decrease in the percentage of complete genes, with 50.2% for *B. jararaca* and 40.2% for *B. asper*. Importantly, BUSCO tends to recover more genes when using the complete genome of the organism, but when using specific tissues, the value tends to decrease [20]. Therefore, using a specific tissue such as the venom gland means that BUSCO does not recover all the information; however, using a broad lineage such as eukaryotes can identify more gene matches, even if not all of them are directly related to the venom gland tissue. Another parameter of assembly quality was the mapping of the reference transcriptome against the raw data, obtaining values of 89.01% for *Bothrops asper* and 89.64% for *Bothrops jararaca*. Assemblies with low percentages of mapped reads (<80%) may indicate technical problems, such as sequencing errors, poor quality reads, or insufficient coverage [21].

### 2.2. Protein Identification

Analysis of the assembled transcriptomes against the non-redundant protein database revealed that SVMPs (snake venom metalloproteinases) and SVSPs (snake venom serine proteinases) are the most abundant toxin transcripts (Figure 3). On the other hand, the percentage of transcripts encoding CTLs (C-type lectins) and PLA2s (phospholipases A2) show a notable variation between *B. asper* and *B. jararaca*. These differences may be due to geographical variability, age, and diet [11,22,23,24]. Other analyses in species of the genus *Bothrops* have shown that the most common toxin groups are metalloproteinases (SVMPs), bradykinin-enhancing peptides (BPPs), serine proteinases (SVSPs), phospholipase A2 (PLA2), and C-type lectins (CTLs), which have a similar pattern to that observed [11,25,26,27].

The proteins of interest identified with a local BLAST of the assembled transcriptome of *Bothrops asper* and *Bothrops jararaca*, showed different degrees of similarity as some sequences expressed more coverage than others. Table 2 details the best E-value obtained in each group of proteins (phospholipase A2, sphingomyelinase D, conotoxin, type B toxins (CfTxB), allergen 5, arylsulfatase I, and three-finger toxins (Sntx 3).

It is worth mentioning that, in some of the results discussed in Section 2.3, sequences with higher coverage were selected even when their E-values were relatively high. The table reflects the best E-values from the BLAST search for each protein; however, some E-values are considerably far from zero. In these cases, sequences were chosen based on their coverage to recover as much information as possible about the novel proteins. Consequently, higher E-values are presented, even though values close to zero are ideal.

### 2.3. Multiple Alignment

#### 2.3.1. Conotoxin

Analysis of the *Bothrops jararaca* and *Bothrops asper* contigs revealed the presence of conserved residues in Chi-conotoxin, the non-cytoplasmic domains and Conotoxin-T.

Proteomic studies of snake venom have focused on highly expressed proteins such as phospholipases A2, metalloproteinases, serine proteinases, and three-finger toxins [28]. In some species, these proteins can constitute up to 80% of the total venom, as seen with phospholipase A2 or metalloproteinases [26,29]. In contrast, proteins in lower proportions characterized the auxiliary functions and had less variability [30]. Despite their lower prevalence, these proteins interact with specific targets in various physiological systems, leading to side effects such as ion channel blockade, blood coagulation disturbances, fibrinolysis, and inflammation [30,31].

This research analyzed toxins from various organisms to identify new proteins in the venom gland transcriptomes of *Bothrops asper* and *Bothrops jararaca*. First, conotoxins are essential in the venoms of snails from the genus *Conus* [32]. Their primary effects target ion channels, G-protein-coupled receptors, and neurotransmitters [33,34]. There are over 100 types of conotoxins with different physiological activities, with the T-superfamily being particularly noteworthy [33]. Their structure, characterized by disulfide bridges and cysteine residues, affects the nicotinic acetylcholine receptor (nAChR), inhibiting neuronal and neuromuscular transmission [33,35]. In the alignment shown in Figure 4, matches for Conotoxin-T and Chi-conotoxin were scarce among the contigs of *B. asper* and *B. jararaca*.

This result was expected, as the conotoxin family is known for its unique structure, which can be specific to each species within the genus *Conus*, facilitating binding to various isoforms of ion channels and affecting different parts of the nervous system [34]. In this context, neuromuscular paralysis is the primary effect associated with conotoxins [36]. On the other hand, while the venom of *Bothrops* snakes can induce paralysis, its main function is to affect cardiovascular functions and hemostatic systems [12,37,38].

#### 2.3.2. CftxB

Analysis of type B toxin (CftxB) showed that it is related to the cytoplasmic domain and membrane regions. Some amino acids of these sequences match the contigs of *Bothrops asper* and *Bothrops jararaca*.

Jellyfish venoms show a wide variability in their activity and composition, with numerous proteins yet to be discovered. Among the species of medical importance is *Chironex fleckeri*, whose fast-acting venom causes damage to the cardiovascular and pulmonary systems [39,40]. Proteomic analyses indicate that the main toxins synthesized by jellyfish are classified as Type I (CfTX-1 and CfTX-2) with hemolytic activity and Type II (CfTX-A and CfTX-B) that generate cardiovascular collapse [41]. These effects occur because the toxins use pore formation, cell membrane collapse, and ion channel modulation to promote ion entry into their prey, increasing calcium levels [41,42]. Among the proteins recovered with Pfam in Figure 5, there are the amino acids of the cytoplasmic domain and the membrane regions, which coincide with the contigs of *Bothrops asper* and *Bothrops jararaca*. These proteins are part of the integral membrane proteins that may be related to snake venom [43]. In viperid venom, the protein family of disintegrin-type metalloproteinase (ADAM) and in C-type lectins, it was identified that their structure expresses a transmembrane domain and a cytoplasmic domain [43,44]. In addition, the structure of ADAM proteins allows a connection with the cellular and extracellular matrix, generating adhesion, migration, proteolysis, and cell signaling [43,44,45]. Because the cytoplasmic domain and membrane regions are present in a wide range of proteins, it is normal to expect this amino acid coincidence with the *Bothrops* species analyzed. It is important to consider that the *B. asper* contig is much larger than that of *B. jararaca*, so a greater amount of information was recovered that largely coincides with the cytoplasmic domain.

#### 2.3.3. Phospholipase A2

The domains and active sites of phospholipase A2 from *Oxyuranus scutellatus*, a snake with neurotoxic venom, conserved amino acids with the contigs of *Bothrops asper* and *Bothrops jararaca*.

An individual analysis of the active sites showed that the sequences are more conserved for *Bothrops asper* and *Bothrops jararaca* compared to *Oxyuranus scutellatus*. Furthermore, the recovered sequences differ in some amino acids and are therefore believed to be species-specific, even within the same genus.

Within the protein arsenal of snake venom, phospholipase A2 (PLA2) has undergone an accelerated evolutionary process; in fact, it is estimated that there are more than 600 varieties of PLA2 [46,47,48]. This protein has adapted to the needs of each type of organism, presenting neurotoxic activity for snakes of the Elapidae family and myotoxic activity for the Viperidae family [48,49,50]. It has been documented that PLA2 isoforms can share between 40% and 99% identity in their amino acid sequences, as in Figure 6, which shows a large number of matches, despite the studied species being different [28,50]. PLA2 is classified into four groups, including group I for elapids such as *Oxyuranus scutellatus* and group II for viperids such as snakes of the genus *Bothrops* [51]. Both share seven disulfide bridges in their structure, along with two highly conserved catalytic residues, histidine (His48) and aspartic acid (Asp99) [28]. These conserved sites are shown in Figure 7. Although certain amino acids of the conserved sites differ in the sequences of *Bothrops asper* and *Bothrops jararaca*, these variations should not interfere, since the positions of the cysteine (Cys) amino acids are maintained. Cysteine is an important amino acid for forming disulfide bonds that provide stability and resistance to denaturation [52]. Importantly, the sequences obtained for *B. asper* and *B. jararaca* were subjected to Blastp analysis on the NCBI web server, revealing similarities with PLA2s from several snake species, but none related specifically to *B. asper* and *B. jararaca*. Therefore, this study presents a new PLA2 isoform for these species.

#### 2.3.4. Sphingomyelinase D

Sphingomyelinase D analysis revealed that the glycerophosphodiester phosphodiesterase (GDPD) domain is only conserved in a specific section within the *Bothrops asper* and *Bothrops jararaca* contigs.

Based on the result in Figure 8, the fragment of the glycerophosphodiester phosphodiesterase domain was investigated, which showed coincidences with the contigs of *Bothrops asper* and *Bothrops jararaca*. This led to the identification of the phosphodiesterase-like phospholipase C (PLC) family and the glycerophosphodiester phosphodiesterase domains, which show differences compared to the sphingomyelinase D of *Loxosceles laeta*, as well as between species of the genus *Bothrops*.

Spiders of the Sicariidae family, such as Loxosceles and Sicarius, cause dermonecrotic skin lesions in humans through the enzyme sphingomyelinase D (SMase D) [53]. Blast analysis showed some relationship between this protein and the venom gland transcriptome of *Bothrops asper* and *Bothrops jararaca* (Figure 8). According to Lee and Lynch [54], SMase D is a phospholipase that can hydrolyze sphingomyelin (SM), as well as a series of glycerophospholipids causing damage to blood vessels, stimulating cell proliferation, endothelial hyperpermeability, and inflammation [55]. In snakes, these effects can be caused by vascular endothelial growth factor (VEGF) or CRISP family proteins [56,57]. Gomes et al. [55] demonstrated that the venom of snakes such as *Bothrops jararaca*, *Crotalus durissus*, *Lachesis muta,* and *Micrurus frontalis* lacks the protein activity of SMase D. This was supported by the multiple alignment in Figure 8, where the matching amino acids are few, preventing the identification of domains and active sites. However, at the end of the alignment, a section of the glycerophosphodiester phosphodiesterase (GDPD) domain of *Loxosceles laeta* matches *B. asper* and *B. jararaca*. This section as studied further in Figure 9, identified for *B. asper* and *B. jararaca*, the phosphodiesterase-like phospholipase C (PLC) family and the glycerophosphodiester phosphodiesterase domains. Phospholipase C (PLC) regulates the transduction of signals, hormones, neurotransmitters, and growth factors [58]. This protein has been widely studied in mammals due to its ability to hydrolyze phosphatidylinositol 4,5-bisphosphate (PIP2), giving rise to inositol 1,4,5-trisphosphate (IP3) and diacylglycerol (DAG), responsible for regulating the cellular signaling mentioned above [59,60]. Costa et al. [61] analyzed the serine protease (SVSP) of the venom of *Crotalus durissus terrificus*, showing that among its receptors is PLC, which is involved in the edema effect through the metabolism of arachidonic acid. Although no information on PLC was found in the databases (NCBI, UNIPROT, KEGG) in species of the genus *Bothrops*, it is expected to be present, since this protein acts together with protease-activated receptors (PARs), which have effects in homeostasis, thrombosis, and inflammation [60,61].

On the other hand, glycerophosphodiester phosphodiesterase (GD-PDE) domains are mainly found in bacteria whose function is to degrade mammalian membranes [62,63]. However, its function in snakes is unknown and the presence of this protein is possibly due to the interaction of bacteria with the venom gland of *Bothrops asper* and *Bothrops jararaca* [64,65,66]. Previous studies have shown that some bacteria have adapted to the venom of snakes, snails, spiders, and scorpions [64,66,67]. Esmaeilishirazifard et al. [66] identified *Enterococcus faecalis* sequences resistant to *Naja nigricollis* venom. The venoms of snakes, spiders, and other animals have been recognized for their antimicrobial activity; however, because of the fangs of venomous snakes exposed to the environment, they are susceptible to bacteria entering and adapting [65,66]. In snails of the genus *Conus*, symbiosis with *Stenotrophomonas* has been reported in the venom ducts, this bacteria contributes with small protease molecules generating variations in the venom [64]. The scarce literature on the oral microbiota of snakes has not been able to fully reveal information about the microbial communities in the venom glands. For this reason, it is still unknown whether the microorganisms identified in these tissues could offer advantages in the toxic composition of the venom [68,69]. In this case, it is suspected that bacteria related to the production of glycerophosphodiester phosphodiesterase and similar domains could have colonized the venom gland of *B. asper* and *B. jararaca*. However, additional studies are required to confirm this hypothesis.

#### 2.3.5. Arylsulfatase I

Pfam analysis of *Novipirellula aureliae* revealed the presence of sulfatase-like active sites and esterase-like SGNH hydrolase domains, as well as SGNH hydrolase families. However, only the SGNH hydrolase family and domain show matches with the contigs of *Bothrops asper* and *Bothrops jararaca*.

Due to the extensive length of the *Novipirellula aureliae* arylsulfatase I sequence, continuity of the SGNH hydrolase family and domain cannot be discerned in the *Bothrops asper* and *Bothrops jararaca* contigs. Therefore, a multiple alignment focusing on the SGNH hydrolase family and domain was performed.

#### 2.3.6. Arylsulfatase I

Analysis of arylsulfatase I from the king cobra *Ophiophagus hannah* revealed sulfatase-like active sites and the B-type arylsulfatase domain, as well as the sulfatase domain and N-terminal region. In contrast to the alignment in Figure 10, the contigs from *Bothrops asper* and *Bothrops jararaca* have much more complete and conserved sequences.

The bacterium *Novipirellula aureliae*, with the accession number TWU36542.1, was isolated from the jellyfish *Aurelia aurita* by Kallscheuer and collaborators in 2019 [70]. The microbiome of this species is one of the most studied within the Ulmaridae family, with cases of endosymbiosis with *Mycoplasma* strains found in the sessile (polyp) life phase of the jellyfish [71]. In addition, bacterial strains associated with aromatic hydrocarbons such as *Stenotrophomonas*, *Pseudomonas*, *Burkholderia*, *Achromobacter,* and *Cupriavidus* were identified [72]. On the other hand, the genus *Novipirellula* has been associated with antimicrobial activity. Among the investigations, Vitorino in 2024 [73] demonstrated that strains of *Novipirellula caenicola* and *Rhodopirellula* spp. have an effect on Gram-positive bacteria. Studies in *A. aurita* identified an antimicrobial peptide, called aurelin, which inhibits Gram-positive and -negative bacteria [74]. Therefore, survival in this environment could be possible only for some bacteria, where possibly *N. aureliae* contributes to the antimicrobial effect; however, this has not yet been confirmed.

Among the proteins found in *N. aureliae*, there is the arylsulfatase with its sulfatase-like active sites and the SGNH hydrolase domain. The latter is the only sequence that fully matches the contigs of *Bothrops asper* and *Bothrops jararaca* (Figure 10). The SGNH family and hydrolase domain are distributed in vertebrates, plants, bacteria, fungi, and archaea, fulfilling functions in cell signaling, biomass conversion, and pathogenesis [75]. Proteins containing this domain have the ability to act as esterases and lipases, in addition to presenting conserved residues such as serine, glycine, asparagine, and histidine, which are also found in phospholipases A2 from snake venom [76,77]. The alignments shown in Figure 11 seem to be conserved in *B. asper* and *B. jararaca*, because SGNH is related to phospholipase A2, and the latter is a protein commonly present in snake venom. On the other hand, arylsulfatase-I from *Ophiophagus hannah* (Figure 12) contains sulfatase-like active sites and the arylsulfatase type B domain. These proteins match the contigs of *Bothrops asper* and *Bothrops jararaca*, obtaining more complete and conserved sequences than those analyzed in Figure 10. It is important to note that the contigs recovered for the analysis of Figure 10 and Figure 12 are completely different; in the first alignment there is more conservation against the esterase-type SGNH hydrolase domains, while in the second the sulfatase domains and the arylsulfatase family are mostly conserved. The protein analyzed in both cases is the same (Arylsulfatase I), however, they are in different organisms causing their function to differ, because their biological needs are completely different. For example, the arylsulfatase in *Novipirellula aureliae* has a highly conserved domain (SGNH) in several lineages carrying out cell signaling, biomass conversion and pathogenesis functions; on the other hand, the arylsulfatase of *O. hannah* has the Arylsulfatase domain which is responsible for eliminating sulfate groups causing tissue degradation.

In the alignment of Figure 12, the presence of arylsulfatase B (ARSB) stands out, a protein that eliminates the 4-sulfate group from sulfated glycosaminoglycans (GAGs) [78,79]. In humans, ARSB regulates physiological processes, and its deficiency leads to myocarditis, lung diseases, neurological disorders, liver dysfunctions, and problems in bones and cartilage [79]. ARSB has also been identified in the venom glands of some animals such as scorpions (*Diplocentrus whitei*), wasps (*Nasonia vitripennis*), centipedes (*Theatops posticus*), and snakes (*Naja nigricolis*) [80,81,82,83]. The venom of *N. nigricolis* contains a wide range of proteins, among which is ARSB, whose effect is attributed to the breakdown of GAGs in connective tissue, ligaments, and tendons, while in *N. vitripennis* a possible action in digestion is considered [81,82]. These may be the first ARSB toxins to be identified in the venom gland transcriptome of *B. asper* and *B. jararaca*, and their effect could be related to the ARSB present in *N. nigricolis*.

The presence of the ARSB toxin in the venom gland transcriptomes of *Bothrops asper* and *Bothrops jararaca* may contribute to variability in venom composition, potentially enhancing some of the previously mentioned effects. Several studies demonstrated that antivenoms do not fully inhibit all toxins present, particularly those found in lower abundance [23,84,85,86]. Due to the chemical diversity of toxins, antivenoms appear to have a more limited scope in inhibiting effects such as neurotoxicity, coagulopathies, myotoxicity, renal failure, and tissue damage [87,88,89,90]. The lack of inhibition of arylsulfatase may be linked to complications related to coagulopathies, which have been reported as secondary effects in viper envenomations, even following treatment [86,91,92]. However, Nok and collaborators [81] demonstrated that arylsulfatase can be inhibited by cations such as Co^2+^ and Mn^2+^, suggesting a potential application for improving the efficacy of antivenom therapies. This opens the possibility of developing more specific antivenoms that, in addition to neutralizing predominant toxins, could include inhibitors targeting less abundant proteins like arylsulfatase. The incorporation of specific inhibitors or the use of combination therapies could optimize the neutralization of toxins that are not immunocaptured, thereby improving treatment and reducing the side effects associated with venom variability.

#### 2.3.7. Sntx-3

Analysis of the three-finger neurotoxin (Sntx 3) identified the active site of the CD59 antigen and the Ly6/PLAUR domain. In addition, a superfamily homologous to the three-finger neurotoxin was found. The contigs of *Bothrops asper* and *Bothrops jararaca* show several conserved amino acids in relation to the domain, superfamily, and active site mentioned above, although Pfam did not detect an active site for *B. asper*. However, there is considerable variation between species of the genus *Bothrops* compared to the neurotoxin of *Oxyuranus scutellatus*.

Three-finger toxins (3FTx) have a wide range of biological activities, with over 700 proteins reported in the UniProt database [93]. Although these toxins have been widely studied in the venom of elapid snakes, which can constitute up to 95% of the venom, they have also been found in nonvenomous snakes [93,94]. The continued evolution of 3FTx toxins has resulted in the dispersal of numerous isoforms in various animals [95], giving rise to proteins with 3FTx-like structures, such as lymphocyte antigen 6 (Ly6) and CD59, identified in this research (Figure 13). It is important to note that the 3FTx toxins present in the venom of elapid snakes originate from a nontoxic protein that underwent genetic duplication and moved away from its plesiotypic or ancestral form [95,96]. However, the 3FTx toxins found in boas and viperids retain the ancestral structure composed of ten cysteines, functioning as α-neurotoxins that are lethal to birds and reptiles but less harmful to mammals [94]. Mackessy [95] points out that 3FTx toxins in viperids have been erroneously called “weak neurotoxins” since they have similar effects to the 3FTx of elapid snakes. The difference is that in vipers, 3FTx is secreted at extremely low levels, as these snakes promote the production of other proteins such as PLA2 or metalloproteinases.

#### 2.3.8. Allergen 5

Analysis of allergen 5 revealed the CRISP domain, the CAP superfamily, and allergen family 5 sequences. *Bothrops asper* and *Bothrops jararaca* contigs show several conserved amino acids relative to the domain and families mentioned above.

Although the CRISP domain, the CAP superfamily, and allergen family 5 sequences were identified, Figure 14 shows amino acid variation in the *Bothrops asper* and *Bothrops jararaca* contigs, so an alignment performed for each species with their respective domains and families was identified in Pfam.

The CAP superfamily is a set of highly conserved proteins found in both prokaryotes and eukaryotes, consisting of three families: cysteine-rich proteins (CRISP), antigen 5 (Ag5), and pathogenesis-related proteins (PR-1) [97]. This superfamily is in the venom of various species, causing inflammatory, proteolytic effects on ion channels and immune regulators [98]. The Ag5 family is widely expressed in insect venom, while the CRISP family is in reptile venom, such as snakes from the Colubridae, Elapidae, and Viperidae families [99]. However, in snake venom, CRISP proteins are less abundant than metalloproteinases and phospholipases A2, as observed in *Bothrops asper* and *Bothrops jararaca*, where their expression is less than 4% [22,26,100]. Due to its low expression level, the biological significance of this protein family in venoms is not entirely clear [99,101]. However, previous studies suggest that snake CRISPs can block smooth muscle contraction and, thanks to their ability to interact with calcium and potassium channels, can associate with other proteins, such as metalloproteinases and phospholipases A2, leading to inflammation and delayed wound healing [98,99,101].

As mentioned above, the CRISP family has been identified in snakes from different families, however, these tendencies vary so the CRISP proteins identified in Figure 15 are specific to *Bothrops asper* and *Bothrops jararaca*. On the other hand, the Ag5 family, which is specific to insects, was also identified. So far, the presence of Ag5 in snakes of the genus *Bothrops* has not been detailed, but the NCBI database contains data on antigens in the genome of some snake genera such as *Protobothrops*, *Naja*, *Crotalus,* and *Ophiophagus*. It is believed that the presence of this protein in the venom gland of *B. asper* and *B. jararaca* could be due to the fact that, as Ag5 belongs to the CAP superfamily, it may share certain relationships with the CRISP family [98,102,103]. In addition, the anticoagulant function of Ag5 could be within the wide repertoire of toxins in the venom of these snakes, although its low expression could make them not so evident.

## 3. Conclusions

Our analysis provides the first assembled venom gland transcriptome for *Bothrops asper* and offers a broader perspective on the venom gland transcriptome of *Bothrops jararaca*. By performing a BLASTp analysis of the matching sequences in the alignments through the NCBI web server, the presence of new proteins that have not been detailed so far in either species, *B. asper* or *B. jararaca*, was revealed. New isoforms of phospholipases, CRISP proteins, and three-finger toxin-like proteins (3FTx) such as CD59 and the Ly6/PLAUR domain were found. This shows once again how the evolutionary process of venom in snakes can be so variable by environment, but there are conserved structures as evidence of an ancestral protein that is still unknown. In addition, new proteins such as glycerophosphodiester phosphodiesterase, antigen-5-related proteins, and arylsulfatases were detailed, whose biological function remains to be detailed. These findings support the idea that we have contributed new knowledge to the understanding of the protein composition of the venom glands of *B. asper* and *B. jararaca*. This research establishes a foundational step in the identification of novel proteins. The next phase involves modeling these proteins to deepen the understanding of their physiological functions. This includes analyzing the folding of their functional motifs and comparing them with other protein models. Such a comprehensive approach will enable a thorough evaluation of their structures and potential functions, allowing for an in-depth exploration of their biological roles and therapeutic implications. Given their potential involvement in coagulopathies, neurotoxic effects, thrombosis, and inflammatory responses, this work could contribute significantly to the development of innovative treatments based on venom components.

## 4. Materials and Methods

### 4.1. Protein Data

The protein sequences belonging to the families of allergen 5, sphingomyelinase D, conotoxin, type A toxins (CfTX-A), phospholipase A2, three-finger toxins, and arylsulfatase, detected in the venom glands of insects (*Vespula germanica*), arachnids (*Loxosceles laeta*), marine gastropod (*Conus ebraeus*), cnidarians (*Chironex fleckeri*), reptiles (*Oxyuranus scutellatus*, *Ophiophagus hannah*), and bacteria (*Novipirellula aureliae*), respectively, and their accession codes are detailed in Table 3.

Proteins included in Table 3 were analyzed in the venom gland transcriptome of *Bothrops jararaca* and *Bothrops asper*.

### 4.2. De Novo Transcriptome Assembly and Quality Analysis

The venom gland transcriptomes of *Bothrops jararaca* and *Bothrops asper*, were obtained from the “SRA” database of NCBI, sequencing by Illumina in paired end format; the access codes are SRR12800503 and SRR12915695, respectively. No biological samples were used, and no procedures were performed on animals.

The venom gland transcriptomes of *Bothrops jararaca* and *Bothrops asper* comprise 16,574,152 and 13,295,995 paired-end reads, respectively. Quality assessment of the raw reads was conducted using FASTQC (v0.11.5) [104], followed by adapter removal and trimming of low-quality sequences (Phred score < 20) using Trimmomatic (v0.39) [105]. Post-cleaning, a total of 4,553,854 reads for *B. jararaca* and 4,026,519 reads for *B. asper* were retained. The de novo assembly was performed with Trinity (v2.15.1) [106], using a default K-mer size of 25, resulting in 76,821 contigs for *B. jararaca* and 54,771 contigs for *B. asper*. [104,105,106]. Statistical analyses, including N50 value, total genes, and transcripts were performed using the Perl script ‘TrinityStats.pl’ from the TrinityRNAseq-v2.15.1 tool package [105]. The assembled transcriptomes were then mapped to the raw data using Bowtie2-v2.3.4.1 [105] to examine coverage.

Transcriptome completeness was assessed with the Benchmarking of Universal Single Copy (BUSCO) sets with the eukaryotic and sauropsid lineages, using the BUSCO-v5.5.0 software version [106].

### 4.3. Functional Annotation and Analysis of Proteins of Interest

The functional annotation of the assembled transcriptomes of the venom gland of *Bothrops asper* and *Bothrops jararaca*, was performed through a BLASTX search with an E value of 1 × 10^−5^ in Diamond-v2.0.9 [107], using the NCBI non-redundant (NR) protein database.

A local database of the assemblies was generated through the makeblastdb command, for detecting the new proteins in the transcriptome of the venom glands of both species. The search for the query proteins was executed using the tblastn option. This approach facilitated the identification of the most matching sequences, selecting those with the best E value (Table 2). Active sites, conserved domains, and protein families were identified in Pfam page. These sequences were useful to perform the multiple alignment with muscle-v3.8.31 [108], which were visualized with Jalview-v2.11.3.2 [109]. Additional data on the multiple alignment of the new proteins can be found in the Appendix A.

## Figures and Tables

**Figure 1 toxins-16-00511-f001:**
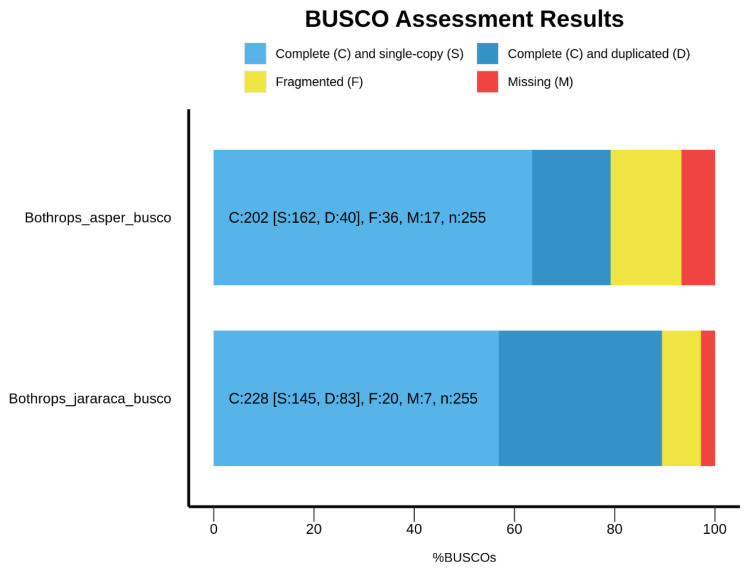
Quality assessment of assembled transcriptomes using BUSCO (Benchmarking Universal Single-Copy Orthologs) analysis with the Eukaryotic lineage_odb10 which shows the percentage of complete genes for *Bothrops asper* and *Bothrops jararaca*.

**Figure 2 toxins-16-00511-f002:**
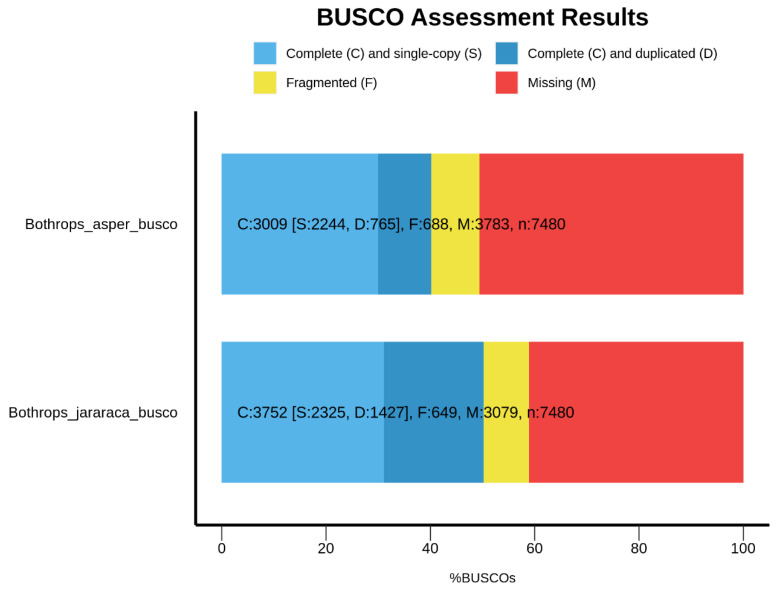
Quality assessment of assembled transcriptomes using BUSCO (Benchmarking Universal Single-Copy Orthologs) analysis with the Sauropsid lineage_odb10 which shows the percentage of complete genes for *Bothrops asper* and *Bothrops jararaca*.

**Figure 3 toxins-16-00511-f003:**
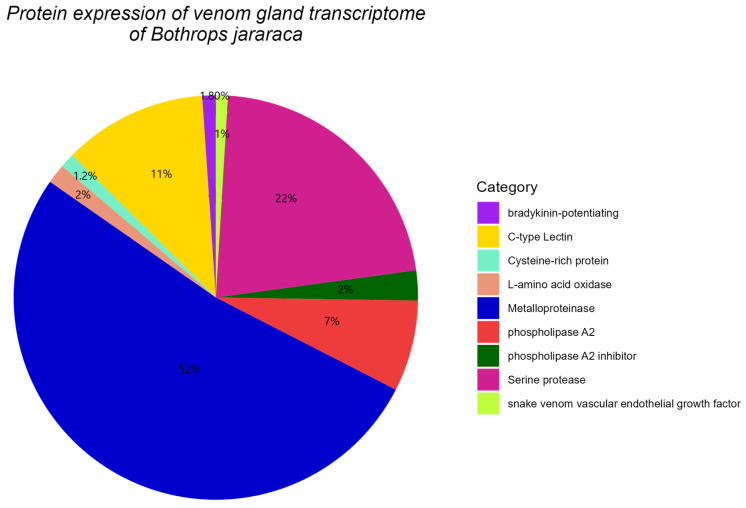
Transcriptomic profile of the venom glands of *Bothrops asper* and *Bothrops jararaca*. Metalloproteinase (SVMP), bradykinin-enhancing peptide (BPP), snake venom serine proteinases (SVSPs), C-type lectin (CTL), vascular endothelial growth factor (svVEGF), phospholipase A2 (PLA2), phospholipase A2 inhibitor (PLA2 inhibitor), cysteine-rich proteins (CRISPs) and L-amino acid oxidases (LAAOs). Novel proteins include arylsulfatase (ARS) and glycerophosphodiester phosphodiesterase (GDPD).

**Figure 4 toxins-16-00511-f004:**
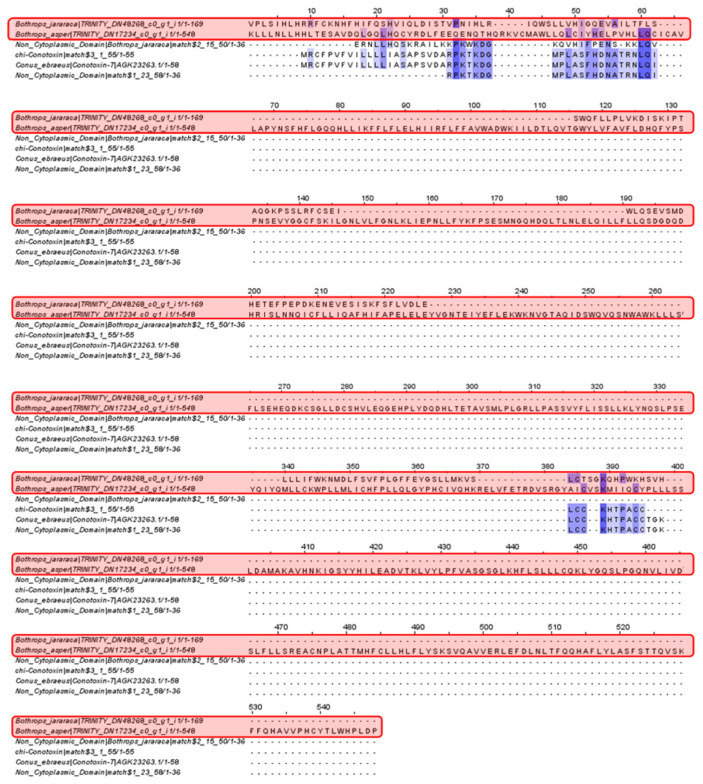
Multiple sequence alignment of conotoxin. *Bothrops asper* and *Bothrops jararaca* contigs identified in the venom gland transcriptome highlighted in red compared to the non-cytoplasmic domain proteins, Chi-conotoxin and Conotoxin-T. Domain sequences were obtained from the Pfam database.

**Figure 5 toxins-16-00511-f005:**
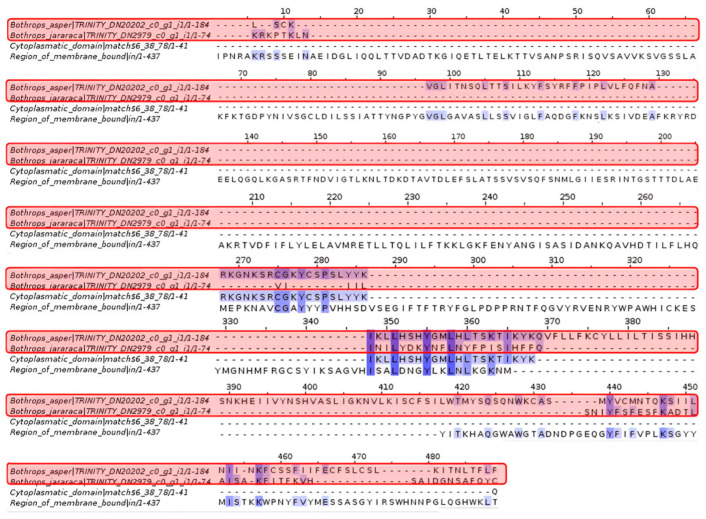
Multiple sequence alignment of type B toxin (CftxB). *Bothrops asper* and *Bothrops jararaca* contigs identified in the venom gland transcriptome highlighted in red compared to proteins in the cytoplasmic domain and membrane regions. Domain sequences were obtained from the Pfam database.

**Figure 6 toxins-16-00511-f006:**
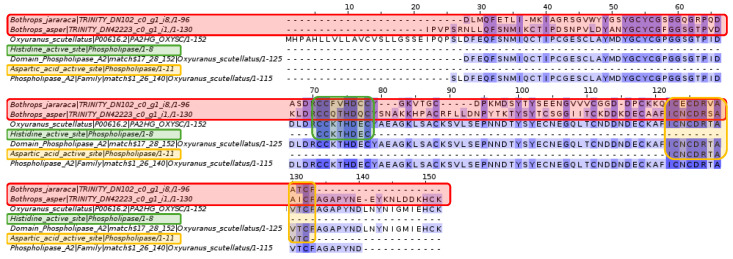
Multiple sequence alignment of phospholipase A2. *Bothrops asper* and *Bothrops jararaca* contigs identified in the venom gland transcriptome highlighted in red compared to proteins of the phospholipase A2 domain and family. The histidine active site is highlighted in green and the aspartic acid active site in orange. Domain sequences were obtained from the Pfam database.

**Figure 7 toxins-16-00511-f007:**
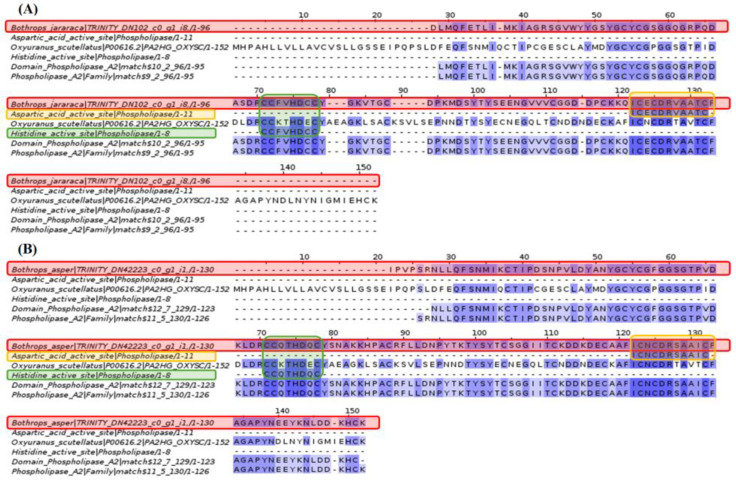
Multiple sequence alignment of phospholipase A2. (**A**) *Bothrops jararaca* and (**B**) *Bothrops asper*. The histidine active site is highlighted in green, the aspartic acid active site in orange and the contigs of the genus *Bothrops* are in red. Domain sequences were obtained from the Pfam database.

**Figure 8 toxins-16-00511-f008:**
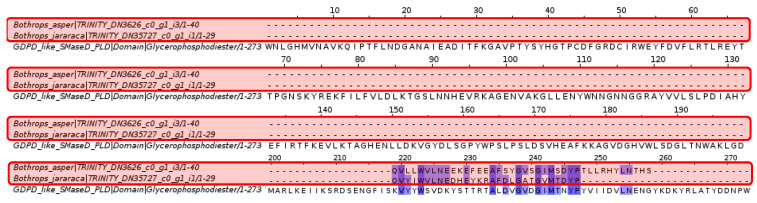
Multiple sequence alignment of sphingomyelinase D. *Bothrops asper* and *Bothrops jararaca* contigs identified in the venom gland transcriptome highlighted in red are compared to sphingomyelinase D-like glycerophosphodiester phosphodiesterase domain (GDPD) proteins. Domain sequences were obtained from the Pfam database.

**Figure 9 toxins-16-00511-f009:**
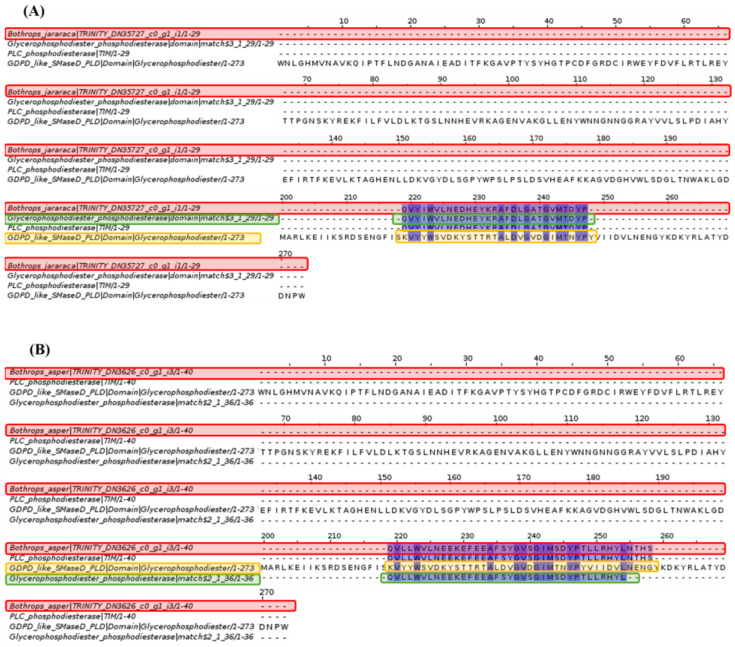
Multiple sequence alignment of sphingomyelinase D. (**A**) *Bothrops jararaca* and (**B**) *Bothrops asper*. The sequences highlighted in red correspond to the species of the genus *Bothrops*, the orange color to the sphingomyelinase D (GDPD) of *Loxosceles laeta,* and the green color are the domains of the glycerophosphodiester phosphodiesterase of *B. jararaca* and *B. asper*. The glycerophosphodiester phosphodiesterase domains are specific for each *Bothrops* species. Domain sequences were obtained from the Pfam database.

**Figure 10 toxins-16-00511-f010:**
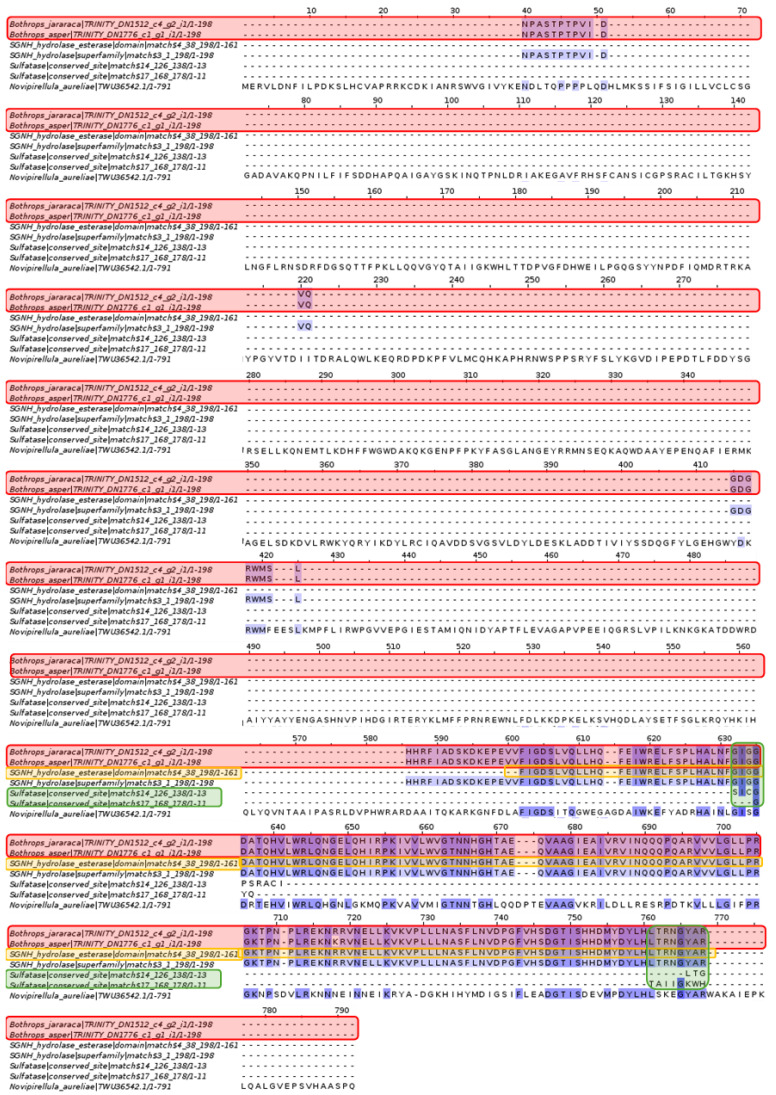
Multiple sequence alignment of arylsulfatase. *Bothrops asper* and *Bothrops jararaca* contigs identified in the venom gland transcriptome are highlighted in red compared to the sulfatase-like active sites in green and SGNH esterase-like hydrolase domains in orange. Domain sequences were obtained from the Pfam database.

**Figure 11 toxins-16-00511-f011:**
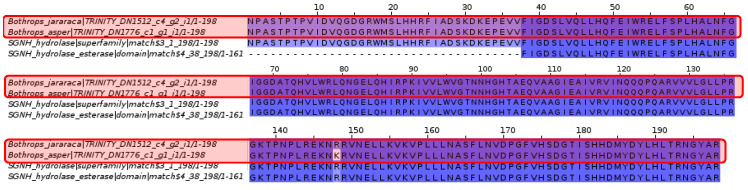
Multiple sequence alignment of arylsulfatase. The contigs of *Bothrops asper* and *Bothrops jararaca* highlighted in red are highly conserved with the SGNH esterase-like hydrolase domain and with the SGNH hydrolase family. Domain sequences were obtained from the Pfam database.

**Figure 12 toxins-16-00511-f012:**
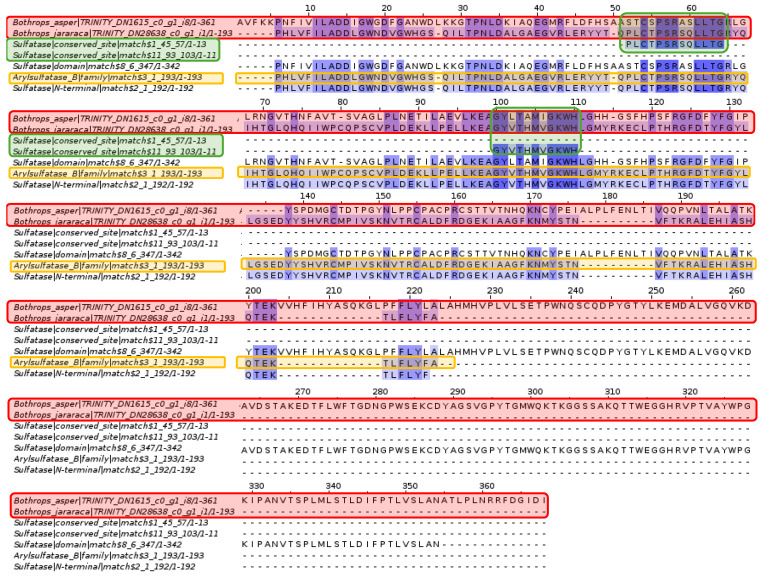
Multiple sequence alignment of arylsulfatase. The *Bothrops asper* and *Bothrops jararaca* contigs, highlighted in red, show high conservation with the esterase-like SGNH hydrolase domain, in green, and the SGNH hydrolase family, in orange. Domain sequences were obtained from the Pfam database.

**Figure 13 toxins-16-00511-f013:**
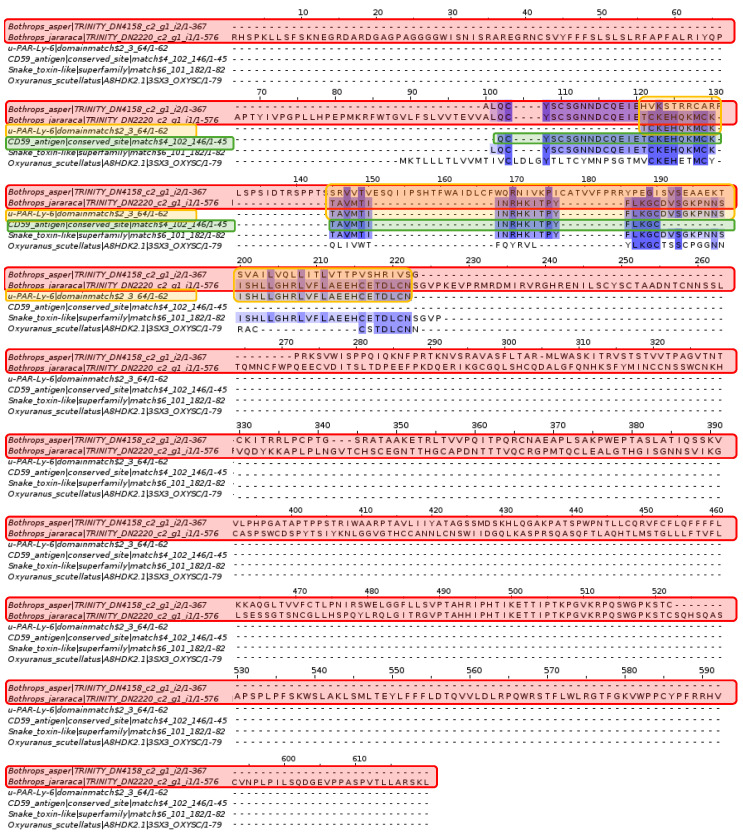
Multiple sequence alignment of Sntx-3. The contigs of *Bothrops asper* and *Bothrops jararaca*, highlighted in red, show high conservation with the Ly6/PLAUR domain, colored orange, and the active site of the CD59 antigen, highlighted in green. The homologous sequence to the three-finger neurotoxin is more conserved in the species of the genus *Bothrops* than with *Oxyuranus scutellatus*. Domain sequences were obtained from the Pfam database.

**Figure 14 toxins-16-00511-f014:**
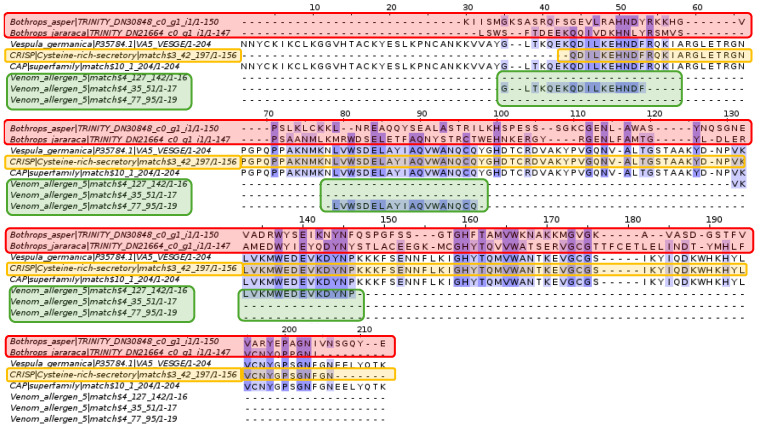
Multiple sequence alignment of allergen 5. *Bothrops asper* and *Bothrops jararaca* contigs, highlighted in red, show conserved amino acids with the CAP superfamily, but more variation with the CRISP domain (orange) and allergen family 5 sequences (green). Domain sequences were obtained from the Pfam database.

**Figure 15 toxins-16-00511-f015:**
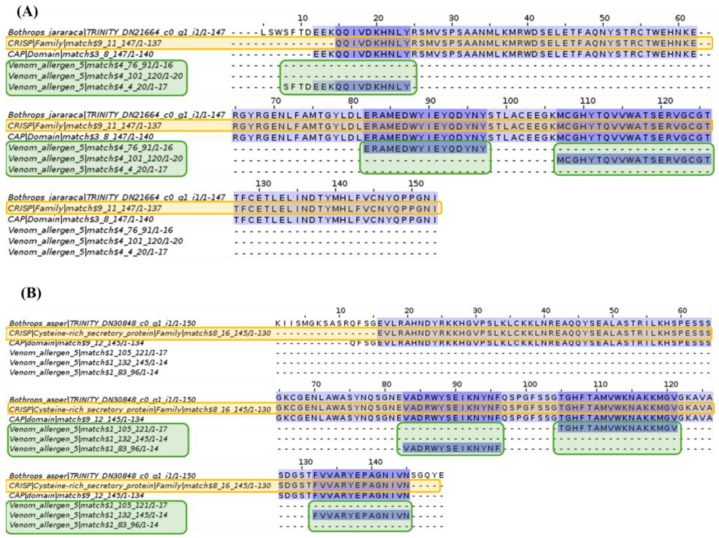
Multiple sequence alignment of sphingomyelinase D. (**A**) *Bothrops jararaca* and (**B**) *Bothrops asper*. Sequences highlighted in orange correspond to the CRISP domain and those in green are sequences from allergen family 5. The proteins are highly conserved for the *B. jararaca* and *B. asper* contigs.

**Table 1 toxins-16-00511-t001:** Assembly quality of the venom gland transcriptomes of *Bothrops asper* and *Bothrops jararaca*.

	*Bothrops asper*	*Bothrops jararaca*
Total number of contigs	13,295,995	16,574,152
Total number of contigs per assembly	54,771	76,821
N10	4629	4198
N20	3300	3301
N30	2504	2667
N40	1948	2152
N50	1453	1678

**Table 2 toxins-16-00511-t002:** E-value of each protein in the assembled transcriptomes of the venom gland of *Bothrops jararaca* and *Bothrops asper*.

Protein	*Bothrops*	E-Value
Conotoxin	*B. asper*	5.4
	*B. jararaca*	1.9
CftxB	*B. asper*	4.5
	*B. jararaca*	5.4
Phospholipase A2	*B. asper*	4 × 10^−51^
	*B. jararaca*	7 × 10^−17^
Sphingomyelinase D	*B. asper*	9.6
	*B. jararaca*	1.3
Glycerophosphodiester phosphodiesterase	*B. asper*	9.6
	*B. jararaca*	1.3
Arylsulfatase I (*Novipirellula aureliae*)	*B. asper*	9 × 10^−34^
	*B. jararaca*	1 × 10^−33^
Arylsulfatase I (*Ophiophagus hannah*)	*B. asper*	9 × 10^−24^
	*B. jararaca*	1 × 10^−74^
Sntx-3	*B. asper*	5.5
	*B. jararaca*	6.4
Allergen 5	*B. asper*	2 × 10^−05^
	*B. jararaca*	1 × 10^−18^

**Table 3 toxins-16-00511-t003:** Proteins from the venom gland of different species.

Protein	Protein Family	Access Code	Species
Arylsulfatase I	Arylsulfatase	TWU36542.1	*Novipirellula aureliae*
Arylsulfatase I	Arylsulfatase	ETE71984.1	*Ophiophagus hannah*
Neurotoxin SNTX-3	Three-finger toxin	A8HDK2.1	*Oxyuranus scutellatus*
Phospholipase A2	Phospholipase A2	P00616.2	*Oxyuranus scutellatus*
Sphingomyelinase D	Phospholipase D	ABD15448.1	*Loxosceles laeta*
Conotoxin	Conotoxin T	AGK23263.1	*Conus ebraeus*
Toxin CfTX-A	Jellyfish toxin	T1PRE3.1	*Chironex fleckeri*
Allergen 5	CRISP	P35784.1	*Vespula germanica*

## Data Availability

The original data presented in the study are openly available in Zenodo repository at https://doi.org/10.5281/zenodo.13821365 (accessed on 7 November 2024).

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
