# Peer review of "Exploring the Venom Gland Transcriptome of Bothrops asper and Bothrops jararaca: De Novo Assembly and Analysis of Novel Toxic Proteins"

_toxins, 2024, doi:10.3390/toxins16120511_

Round 1
Reviewer 1 Report
Comments and Suggestions for Authors
Upon reviewing the article "Exploring the venom gland transcriptome of Bothrops asper and Bothrops jararaca," I have identified several points that need correction both in terms of grammar, missing references and methodological lack of information. Below are the observations:
Line 8: "other proteins appears" - It should be "other proteins appear"
Line 10: "and performed functional annotation" - It should be "and we performed functional annotation" for clarity.
Line 16-17: "These results suggest that the identified proteins may contribute to venom toxic variation and provide new opportunities for antivenom research." Add evidence or references explaining how these proteins are linked to venom variation and how they might influence antivenom development.
Line 57: "many venom proteins to be discovered" - It would be clearer to say "many venom proteins are yet to be discovered."
Line 129-130: "These differences may be due to geographical variability, age and diet." Missing references or data are provided to support this statement.
Line 150-151: "In some results the sequence coverage was considered above the E value." Please explain
Line 158: "the presence of conserved residues" - A clearer structure would be "revealed the presence of conserved residues."
Line 159-160: "Matches for Conotoxin-T and Chi-conotoxin were scarce among the contigs of B. asper and B. jararaca." The scarcity of matches should be better explained or supported with references or comparative data, detailing why these results were expected or unexpected based on prior research.
Line 268: "this allowed the identification of the phosphodiesterase-like" - This can be restructured as "This led to the identification of phosphodiesterase-like..." for a smoother flow.
Line 311-312: "However, its function in snakes is unknown and the presence of this protein is possibly due to the interaction of bacteria with the venom gland of Bothrops asper and Bothrops jararaca."
This hypothesis about bacterial interaction should be elaborated with references to previous studies on bacterial presence in venom glands.
Line 469-470: "It is believed that the presence of this protein in the venom gland of B. asper and B. jararaca could be due to the fact that, as Ag5 belongs to the CAP superfamily, it may share certain relationships with the CRISP family." no references are provided to support the proposed connection between Ag5 and CRISP proteins.
Missing Information in the Methodology:
Im not sure if the authors performed a transcriptome or use previous data from the lab or available data from NCBI. It should be clearer:
If the authors performed a new transcriptome this should be addressed:
Source of Biological Samples: In the methodology section (4.2 and 4.3), there is no clear description of how the venom gland samples of Bothrops asper and Bothrops jararaca were obtained. It is essential to include details on the sample collection process, conditions of sacrifice or anesthesia for the animals, number of specimens used and their origin (e.g., from a laboratory, captured in the wild), ethical approvals and guidelines followed for the use of animals in research.
Sequencing Quality: While FASTQC and Trimmomatic software are mentioned for quality evaluation (lines 509-511), details on the quality parameters applied are missing. Specifically, quality threshold criteria for excluding reads (e.g., minimum quality score), how many reads were removed after quality correction?
Comments on the Quality of English Language
NA
Author Response
Comments 1: Line 8: "other proteins appears" - It should be "other proteins appear"
Response 1: Thank you for pointing this out. We agree with this comment. Therefore, we have made the change on page 1, line 11.
Comments 2: Line 10: "and performed functional annotation" - It should be "and we performed functional annotation" for clarity.
Response 2: Thank you for pointing this out. We agree with this comment. Therefore, we have made the change on page 1, line 14.
Comments 3: Line 16-17: "These results suggest that the identified proteins may contribute to venom toxic variation and provide new opportunities for antivenom research." Add evidence or references explaining how these proteins are linked to venom variation and how they might influence antivenom development.
Response 3: Thank you for pointing this out. We agree with this comment. Therefore, we have made the change on page 15-16, line 405-421.
Comments 4: Line 57: "many venom proteins to be discovered" - It would be clearer to say "many venom proteins are yet to be discovered."
Response 4: Thank you for pointing this out. We agree with this comment. Therefore, we have made the change on page 2, line 58.
Comments 5: Line 129-130: "These differences may be due to geographical variability, age and diet." Missing references or data are provided to support this statement.
Response 5: Thank you for pointing this out. We agree with this comment. Therefore, we have made the change on page 5, line 130.
Comments 6: Line 150-151: "In some results the sequence coverage was considered above the E value." Please explain
Response 6: Thank you for pointing this out. We agree with this comment. Therefore, we have made the change on page 6, line 150-155.
Comments 7: Line 158: "the presence of conserved residues" - A clearer structure would be "revealed the presence of conserved residues."
Response 7: Thank you for pointing this out. We agree with this comment. Therefore, we have made the change on page 6, line 162.
Comments 8: Line 159-160: "Matches for Conotoxin-T and Chi-conotoxin were scarce among the contigs of B. asper and B. jararaca." The scarcity of matches should be better explained or supported with references or comparative data, detailing why these results were expected or unexpected based on prior research.
Response 8: Thank you for pointing this out. We agree with this comment. Therefore, we have made the change on page 7-8, line 187-192. In the paragraph, we explain that these results were expected, given that the conotoxin family is known for its unique structure, which may be species-specific within the genus Conus, affecting different areas of the nervous system. In contrast, the primary function of snake venom is to impact the cardiovascular and hemostatic systems. This explains why there are few coincidences, since conotoxin is highly variable and has a different function than snake venoms.
Comments 9: Line 268: "this allowed the identification of the phosphodiesterase-like" - This can be restructured as "This led to the identification of phosphodiesterase-like..." for a smoother flow.
Response 9: Thank you for pointing this out. We agree with this comment. Therefore, we have made the change on page 10, line 272.
Comments 10: Line 311-312: "However, its function in snakes is unknown and the presence of this protein is possibly due to the interaction of bacteria with the venom gland of Bothrops asper and Bothrops jararaca." This hypothesis about bacterial interaction should be elaborated with references to previous studies on bacterial presence in venom glands.
Response 10: Thank you for pointing this out. We agree with this comment. Therefore, we have made the change on page 12, line 314.
Comments 11: Line 469-470: "It is believed that the presence of this protein in the venom gland of B. asper and B. jararaca could be due to the fact that, as Ag5 belongs to the CAP superfamily, it may share certain relationships with the CRISP family." no references are provided to support the proposed connection between Ag5 and CRISP proteins.
Response 11: Thank you for pointing this out. We agree with this comment. Therefore, we have made the change on page 18, line 490.
Comments 12: Missing Information in the Methodology:
Im not sure if the authors performed a transcriptome or use previous data from the lab or available data from NCBI. It should be clearer.
Response 12: Thank you for pointing this out. We agree with this comment. Therefore, we have made the change on page 19-20, line 530-533. For this study, we used publicly available transcriptome data from NCBI (accessions SRR12800503 for Bothrops asper and SRR12915695 for Bothrops jararaca). No procedures on animals were performed in this study.
Comments 13: Sequencing Quality: While FASTQC and Trimmomatic software are mentioned for quality evaluation (lines 509-511), details on the quality parameters applied are missing. Specifically, quality threshold criteria for excluding reads (e.g., minimum quality score), how many reads were removed after quality correction?
Response 13: Thank you for pointing this out. We agree with this comment. Therefore, we have made the change on page 20, line 534-544.
Acknowledgment: We sincerely appreciate the time and effort the reviewers have invested in reading and providing valuable feedback on our manuscript, which has greatly enhanced the clarity and quality of our work. We are also open to any potential collaborations and welcome further questions or clarifications.
Reviewer 2 Report
Comments and Suggestions for Authors
I am grateful for your submission of the manuscript, entitled 'Exploring the venom gland transcriptome of Bothrops asper and Bothrops jararaca: de novo assembly and analysis of novel toxic proteins'. Following a detailed examination, it has been determined that the study requires further enhancement and refinement in certain aspects before it can be deemed suitable for consideration. Additionally, the current data and analyses are inadequate to substantiate the title and scientific question presented. The following are specific suggestions and comments that were made during the review process, which we hope will prove useful in guiding future revisions.
1. Firstly, although the article employed the BUSCO test to demonstrate a substantial number of homologous single-copy sequences, it did not conduct further analysis of single-copy homologues, such as Orthofindr or Orthomcl. It may be beneficial to investigate these homologous single-copy genes in order to gain a deeper understanding of the evolutionary relationships between these toxins and enzymes and other proteins.
Secondly, the two pie charts in Figure 3 are not readily comparable and analysable. It may be more advantageous to combine them into a single chart. Additionally, the pie chart in question only compares species. It would be beneficial to further analyse the similarities and differences between the two metabolic patterns through enrichment analysis.
3. The multiple sequence alignment plots that follow should be viewed in conjunction with the previous point. Given that Trinity was used, and that Trinity has been shown to have some poor performance in terms of errors, it would have been beneficial for the authors to validate the accuracy and completeness of the splicing using bowtie2. This would have ensured the reliability of the results of the alignment. The multiple sequence comparison plots in the article (e.g. Phospholipase A2, Allergen 5, etc.) are of great importance in demonstrating the conserved and functional domains of the newly discovered proteins. However, the current presentation may lack sufficient intuitiveness, which could result in an increase in the number of sequences as well as motif analysis.
4. The article makes mention of the conservation of the recently discovered venom proteins in a number of different snake species, but lacks the detailed evolutionary analyses that would be required to fully understand the subject matter. It is proposed that phylogenetic trees of the newly discovered venom proteins can be constructed using maximum likelihood or Bayesian methods to analyse the evolutionary relationships of these proteins in different snake species and explore their origins and evolutionary pathways. Alternatively, these newly discovered venom proteins can be compared with known homologues of other snake species (e.g. other members of the Viperidae) to explore the functional differences and adaptive evolution of these proteins in different snake species.
Author Response
Comments 1: Firstly, although the article employed the BUSCO test to demonstrate a substantial number of homologous single-copy sequences, it did not conduct further analysis of single-copy homologues, such as Orthofindr or Orthomcl. It may be beneficial to investigate these homologous single-copy genes in order to gain a deeper understanding of the evolutionary relationships between these toxins and enzymes and other proteins.
Response 1: Thank you for pointing this out. We appreciate your comments and suggestions. Regarding your observation on the use of tools like Orthofindr or OrthoMCL for the analysis of single-copy orthologous genes, we would like to clarify that, while these tools are indeed very useful for studying evolutionary relationships between orthologous genes, the purpose of using BUSCO in our study was to assess the quality of the transcriptome assembly. Our research focuses specifically on the identification and characterization of novel proteins in the venom gland transcriptome, rather than on the in-depth analysis of orthologous gene relationships. Nevertheless, we agree that your proposed analysis could be a highly interesting direction for future research.
Comments 2: Secondly, the two pie charts in Figure 3 are not readily comparable and analysable. It may be more advantageous to combine them into a single chart. Additionally, the pie chart in question only compares species. It would be beneficial to further analyse the similarities and differences between the two metabolic patterns through enrichment analysis.
Response 2: Thank you for pointing this out. We appreciate your comments and suggestions. The pie charts in Figure 3 are intended to highlight variations in protein composition between the venom glands of Bothrops asper and Bothrops jararaca, showcasing differences in protein proportions within the same genus. While we acknowledge the importance of analyzing the metabolic processes of these toxins, this is beyond the scope of the present study. However, we recognize its relevance and will consider it in future research, as metabolic analysis would provide valuable insights into protein functionality and interactions.
Comments 3: The multiple sequence alignment plots that follow should be viewed in conjunction with the previous point. Given that Trinity was used, and that Trinity has been shown to have some poor performance in terms of errors, it would have been beneficial for the authors to validate the accuracy and completeness of the splicing using bowtie2. This would have ensured the reliability of the results of the alignment. The multiple sequence comparison plots in the article (e.g. Phospholipase A2, Allergen 5, etc.) are of great importance in demonstrating the conserved and functional domains of the newly discovered proteins. However, the current presentation may lack sufficient intuitiveness, which could result in an increase in the number of sequences as well as motif analysis.
Response 3: Thank you for pointing this out. We appreciate your comments and suggestions. We used Bowtie2 to assess assembly coverage against the pre-assembly raw data, yielding alignment rates of 89.01% for Bothrops asper and 89.64% for Bothrops jararaca. These percentages suggest that the assembly is continuous and not fragmented. Splicing validation is typically performed at the genomic level, which is beyond the scope of this study, as we focused on transcriptomic data with relatively short recovered sequences, where such analysis is not necessary. Nevertheless, we appreciate the suggestion and will consider it if we expand our work to include genomic analyses in the future.
Comments 4: The article makes mention of the conservation of the recently discovered venom proteins in a number of different snake species, but lacks the detailed evolutionary analyses that would be required to fully understand the subject matter. It is proposed that phylogenetic trees of the newly discovered venom proteins can be constructed using maximum likelihood or Bayesian methods to analyse the evolutionary relationships of these proteins in different snake species and explore their origins and evolutionary pathways. Alternatively, these newly discovered venom proteins can be compared with known homologues of other snake species (e.g. other members of the Viperidae) to explore the functional differences and adaptive evolution of these proteins in different snake species.
Response 4: Thank you for your thoughtful comment regarding the phylogenetic analysis of the newly discovered venom proteins. We appreciate your suggestions. Due to the novelty of these proteins, there are currently limited records of homologous sequences in other snake species, which poses a challenge for constructing a comprehensive phylogenetic tree at this stage. However, we acknowledge the value of this analysis and plan to undertake it in future research as more homologous proteins are identified and additional data become available.
Acknowledgment: We sincerely appreciate the time and effort you dedicated to reviewing our article. Your suggestions have highlighted valuable areas for future research that we aim to explore at the genomic level in Bothrops asper and Bothrops jararaca. However, as of now, no assembled genome for Bothrops asper is available in open-access databases. We hope to begin by obtaining and analyzing genomic data, which will allow us to delve deeper into orthologous relationships, splicing accuracy, and other aspects raised in your feedback. Should the opportunity arise for collaboration, we would be more than willing to pursue it. Please feel free to contact us with any further observations.
Reviewer 3 Report
Comments and Suggestions for Authors
I read the MS entitled: " Exploring the venom gland transcriptome of Bothrops asper and Bothrops jararaca: de novo assembly and analysis of novel toxic proteins", submitted for publication to Toxins.
The MS described a study aimed to identify novel toxic proteins of the venom of Botrops jararaca and Botrops asper, using bioinformatic tools. While other similar studies were performed for B.jararaca, this is the first assembly of the B.asper venom gland transcriptome. Proteins with novel functions potentially implicated for venom toxicity were identified. The results suggest an evolutionary process of venom in snakes driven by the environment, originating from a still unknown ancestral protein. The identification of novel proteins in the venom gland transcriptome of B.asper and B.jararaca, once understood their physiological activity, could be important in the view of their use from a pharmacological point of view.
The paper is well organized and developed. The results of the study are also well described with a deep comparison to previous data and explanation of possible physiological effects of the novel proteins based on literature. However, this very careful explanation of the results is, in my opinion, not followed by equally adequate conclusions. Reading the "Conclusions" section, after a long Results section, I feel like something is lacking or that what was written before is reduced to only few sentences with the first and last two sentences substantially not adding so much. I would have described something more about the evolutionary process, the ancestral unknown precursor. I would also know more about the future steps for understanding the physiological functions of the novel proteins (I'm a little bit afraid of the sentence: "... these proteins will be modeled..... to understand their physiological functions and possible uses in pharmacology..."), and which could be the pharmacological fields of their use. The multidisciplinary nature of this Journal has readers from different fields who can potentially be interested in reading this paper.
Comments on the Quality of English Language
I would make the text more fluent, especially in the Results section through text editing with cuts and summaries
Author Response
Comments 1: The MS described a study aimed to identify novel toxic proteins of the venom of Botrops jararaca and Botrops asper, using bioinformatic tools. While other similar studies were performed for B.jararaca, this is the first assembly of the B.asper venom gland transcriptome. Proteins with novel functions potentially implicated for venom toxicity were identified. The results suggest an evolutionary process of venom in snakes driven by the environment, originating from a still unknown ancestral protein. The identification of novel proteins in the venom gland transcriptome of B.asper and B.jararaca, once understood their physiological activity, could be important in the view of their use from a pharmacological point of view.
The paper is well organized and developed. The results of the study are also well described with a deep comparison to previous data and explanation of possible physiological effects of the novel proteins based on literature. However, this very careful explanation of the results is, in my opinion, not followed by equally adequate conclusions. Reading the "Conclusions" section, after a long Results section, I feel like something is lacking or that what was written before is reduced to only few sentences with the first and last two sentences substantially not adding so much. I would have described something more about the evolutionary process, the ancestral unknown precursor. I would also know more about the future steps for understanding the physiological functions of the novel proteins (I'm a little bit afraid of the sentence: "... these proteins will be modeled..... to understand their physiological functions and possible uses in pharmacology..."), and which could be the pharmacological fields of their use. The multidisciplinary nature of this Journal has readers from different fields who can potentially be interested in reading this paper.
Response 1: Thank you for pointing this out. We agree with this comment. Therefore, we have made the change on page 19, line 509-518. We acknowledge that this study serves as a foundational step in identifying novel proteins, and we intend to further model these proteins to enhance our understanding of their physiological functions. Specifically, we will analyze the folding of their functional motifs and compare them with other protein models. This approach will facilitate a thorough evaluation of their structures and potential functions, allowing us to explore more deeply the biological roles and therapeutic implications of these proteins.
Acknowledgment: We sincerely appreciate your thorough review and insightful observations regarding our manuscript. Your feedback has been invaluable in enhancing the clarity and depth of our conclusions. We are grateful for your suggestions, which will undoubtedly improve the overall quality of our work. Additionally, we are open to clarifying any questions you may have or participating in potential collaborations. Thank you for your time and consideration.
Round 2
Reviewer 1 Report
Comments and Suggestions for Authors
The article is suitable for publication.
Reviewer 2 Report
Comments and Suggestions for Authors
The authors have make great effort to answer the questions ans mend the manuscript. The quality of the manuscript has been improved. I think it is now acceptable for Toxins. I have no more specific comments.